# Appraising the Welfare of Thoroughbred Racehorses in Training in Queensland, Australia: The Incidence, Risk Factors and Outcomes for Horses after Retirement from Racing

**DOI:** 10.3390/ani11010142

**Published:** 2021-01-11

**Authors:** Kylie L. Crawford, Anna Finnane, Ristan M. Greer, Clive J. C. Phillips, Solomon M. Woldeyohannes, Nigel R. Perkins, Benjamin J. Ahern

**Affiliations:** 1School of Veterinary Science, The University of Queensland, Gatton 4343, Australia; s.woldeyohannes@uq.edu.au (S.M.W.); n.perkins1@uq.edu.au (N.R.P.); b.ahern@uq.edu.au (B.J.A.); 2School of Public Health, The University of Queensland, Herston 4006, Australia; a.finnane@uq.edu.au; 3Torus Research, Bridgeman Downs 4035, Australia; rmg@torusresearch.com.au; 4School of Medicine, The University of Queensland, Herston 4006, Australia; 5Curtin University Sustainability Policy (CUSP) Institute, Curtin University, Perth 6845, Australia; Clive.Phillips@curtin.edu.au

**Keywords:** racehorse, thoroughbred, welfare, retirement repurpose

## Abstract

**Simple Summary:**

Whilst the fate of horses after racing has received widespread attention, there is little research into this field. A recent independent inquiry in Queensland, Australia, highlighted that the true outcomes for horses after retirement from racing are largely unknown. This study investigated the retirement of racehorses over a 13-month period. It was found that 0.4% of horses in training per week were retired, and the season and training track did not affect this proportion. The decision for retirement was involuntary, whereby musculoskeletal injuries, respiratory or cardiac conditions and behavioural problems prevented the horse from racing in 56/110 horses (51%). Musculoskeletal injuries were the most common reason for retirement (40/110 horses, 36%). Medium-term follow-up (mostly 14 months, range 8–21) revealed that most horses (108/110; 98%) were repurposed after retirement, almost half as performance horses (50/110; 46%). Horses that were voluntarily retired (retired due to racing form or an impending injury), had 2.28 times the odds of being repurposed as performance horses than those retired involuntarily (*p* = 0.03). There was no association between voluntary or involuntary retirement and whether horses were used for breeding or pleasure. There is a need for traceability and accountability for these horses to ensure that their welfare is maintained in their new careers.

**Abstract:**

There is international public concern regarding retirement of racehorses, including the reason for retirement and the outcome for horses after racing. However, there are currently no prospective studies investigating these factors. A recent independent inquiry in Queensland, Australia, highlighted that the true outcomes for horses after retirement from racing are largely unknown. Furthermore, there are currently no measures to monitor the outcome for racehorses and their welfare once they have left the care of the trainer. This study investigated these gaps in knowledge through a weekly survey conducted over a 13-month period. We aimed to evaluate: (1) the incidence of retirement, (2) the reasons and risk factors for retirement and (3) the medium-term (greater than 6 months) outcomes for horses after retirement. Data were collected through personal structured weekly interviews with participating trainers and analysed using negative binomial and logistic regression. There was a low incidence of retirements, namely 0.4% of horses in training per week. The season and training track did not affect the incidence of retirement. Musculoskeletal injuries were the most common reason for retirement (40/110 horses, 36%). Involuntary retirements accounted for 56/100 (51%) of retirements, whereby musculoskeletal injuries, respiratory or cardiac conditions and behavioural problems prevented the horse from racing The odds of voluntary retirement, whereby the horse was retired due to racing form or impending injury, increased with each additional race start (OR 1.05; *p* = 0.01) and start/year of racing (OR 1.21; *p* = 0.03) but decreased with increasing percentage of first, second and third places (OR 0.94; *p* < 0.001). Medium-term follow-up (median 14 months, IQR 11, 18, range 8–21) revealed that most horses (108/110; 98%) were repurposed after retirement, almost half as performance horses (50/110; 46%). Horses that voluntarily retired had 2.28 times the odds of being repurposed as performance horses than those retired involuntarily (*p* = 0.03). Whether retirement was voluntary or involuntary did not influence whether horses were used for breeding or pleasure. The primary limitation of this study is that our results reflect retirement in racehorses in South East Queensland, Australia, and may not be globally applicable. Furthermore, we were unable to monitor the long-term outcome and welfare of horses in their new careers. It is vital that the industry is focused on understanding the risks for voluntary rather than involuntary retirement and optimising the long-term repurposing of horses. There is a need for traceability and accountability for these horses to ensure that their welfare is maintained in their new careers.

## 1. Introduction

There is international public concern regarding retirement of racehorses, the reason for their retirement and the outcome for these horses after racing; however, there is currently limited research available regarding these topics [1,2,3,4]. Previous studies report that most retirements in New Zealand [5] and the UK [4,6] were voluntary, whereby no injury or condition interfered with training. When retirement was involuntary, musculoskeletal injuries (MSI) were the most common reason, and others included upper and lower respiratory conditions and other miscellaneous conditions including cardiac abnormalities and behavioural problems [5]. There are currently no reports on the incidence or the reasons for retirement of racehorses in Australia.

There are no prospective reports on the outcomes for horses after retirement from racing. One study reported results from a trainer-completed survey regarding the destination of the last five horses to leave their stable [3]. However, many horses in this study did not actually retire from racing; rather, they were moved to other trainers. A recent independent inquiry into the management of retired horses in Queensland highlighted the absence of data regarding the retirement outcomes for racehorses [1]. The true outcomes for horses after retirement from racing are largely unknown due to poor compliance by owners and trainers with requirements to notify control bodies of the retirement or death of racing horses [1]. This situation is exacerbated by the fact that control bodies can be notified of the retirement of a horse, through stable returns, without receiving details of the horse’s outcome [1]. Furthermore, there are currently no measures to follow the outcomes for horses and ensure their welfare of horses once they have left the care of the trainer. The incidence of retirements, the reasons for retirement and the long-term outcomes for these horses are not available and are of great public interest. There is a need for these data to ensure that the welfare of these horses is upheld and to provide public confidence in the transparency of the racing industry.

This study aims to address these gaps in knowledge through a prospective survey conducted over a 13-month period. Our aims were to determine (1) the incidence of retirement, (2) the reasons and risk factors for retirement and (3) the medium-term outcomes for horses after retirement from racing.

## 2. Materials and Methods

### 2.1. Recruitment of Study Participants

This study was conducted concurrently with an investigation into musculoskeletal injuries in South East Queensland thoroughbred racehorses, which describes full details of the study participants and design [7]. Human and animal ethics approvals were obtained from the University of Queensland prior to commencement of this research (approval numbers 2017001248, SVS/384/17, respectively). Thoroughbred racing in Queensland, Australia, is divided into eight regions (Appendix A). Each region is divided into Totalisator Agency Board (TAB) and non-TAB racing clubs. TAB races are televised, and national wagering is available. Non-TAB race meetings are not televised and provide only on-course gambling. Metropolitan (city) TAB race meetings have the greatest prize money awarded, followed by provincial (intermediate) and then regional (country) race meetings. Trainers based at the three metropolitan racetracks under the Brisbane Racing Club (BRC) were invited to participate in our study. Trainers from BRC exercised their horses at either Eagle Farm, Doomben, Deagon, or both Doomben and Eagle Farm, the major racetracks in South East Queensland.

Recruitment of horses was performed by inviting all licenced trainers from BRC with three or more horses in work at the time of recruitment were invited to participate by the first author. A minimum of three horses was selected to ensure efficient data collection and trainer capacity to supply sufficient horses for our concurrent investigation into musculoskeletal injuries. Participating trainers were required to report on all horses under their care.

### 2.2. Study Design

A weekly survey with the first author completing in-person structured interviews with participating trainers or their forepersons regarding retirement of horses was conducted over a 13-month period (56 weeks). Full details of the structured interviews are provided in Appendix A. Structured personal interviews ensured complete data collection and also enabled clarification of any inconsistencies observed. Occasionally, trainers or their forepersons were not available for their scheduled in-person interviews, and in this case, the interview was conducted by telephone, or rescheduled to within seven days. If clarification was required regarding a veterinary diagnosis, data were cross-checked with attending veterinarians. This situation was rare, as most trainers had an excellent understanding of their veterinarians’ diagnoses based on preliminary cross-checks.

### 2.3. Data Collection

Our outcomes of interest included (1) the total number of horses in training for that week and therefore at risk of retirement, (2) the number and details of horses retired from racing, (3) the reason and risk factors for retirement from racing and (4) the medium-term outcome after retirement from racing, whereby medium-term was defined as greater than six months.

### 2.4. Total Number of Horses in Training

We defined a horse in training as a thoroughbred racehorse registered with the Australian Stud Book or equivalent international organization. At the time of the interview, the horse must have been participating in race training under the care of the licenced trainer for a minimum of 5 days a week. These horses were able to train and race without obvious lameness or medical conditions. The total number of horses each trainer had were collected every week. The unit of time at risk was one week. Count (de-identified) data, whereby the number of horses in training, rather than individual horses’ details, were collected.

### 2.5. Number of Horses Retired from Racing

Retirement from racing was defined as the owner’s decision to end the horse’s racing career based on the trainer’s recommendation. This definition did not capture injured horses that were intended to return to racing but for various reasons did not. Details for each retired horse were collected, and retirement was confirmed by cross-checking with the Racing Australia (RA) public database [8]. The reason for retirement and whether the trainer declared behavioural problems existed were also recorded.

### 2.6. Reason for Retirement from Racing

The reason for retirement was obtained from the trainer or foreperson. Reasons were classified into involuntary and voluntary retirements. Reasons for involuntary retirement included the horse having (1) a musculoskeletal injury (MSI) causing an inability to train or race, (2) an upper respiratory condition, (3) a lower respiratory condition, (4) a cardiac problem or (5) a behavioural problem precluding race training including poor barrier manners. Reasons for voluntary retirement included (1) lack of ability, whereby the horse lacked the ability to continue training and racing successfully; (2) reached their level, whereby the horse had performed well up to the current level but lacked the ability to progress to higher classes of races, or (3) impending MSI, whereby the horse had warning signs of MSI (joint effusion, pain on flexion, etc.) that could progress to a clinical problem with continued training and racing.

### 2.7. Risk Factors for Retirement from Racing

The track the horse was trained on, the season that the retirement occurred in and racing career and performance indices were recorded. These potential risk factors could be useful for developing intervention strategies. Racing career and performance indices were obtained or calculated from each individual horse’s form on the RA public database [8]. Career indices included the number of race starts as a two-year-old, age at first race, length of career (months), number of starts, number of starts per 12 months of racing, total race distance accumulated (the sum of the cumulative distance of all races completed in 5 furlongs/1 km units) and the mean distance per start (furlongs). Performance indices included the number and percentage of races won; the number and percentages of races the horse placed 1st, 2nd or 3rd in; the total prize money earned ($1000 AUD), and the prizemoney earned per race start ($1000 AUD). All horses in the study were from Australia or New Zealand, and no horses raced outside of Australia during this study.

### 2.8. Medium-Term Outcome after Retirement from Racing

Every retired horse was followed for 8 to 21 months immediately after retirement. Information on the outcome of the horse was obtained through personal interview with either the trainer, the trainers’ representative or the new owners of the horse, through the trainer. Details of the interview are described in Appendix A. Data collected included, firstly, whether the horse was still alive and, if not, the cause and date of death. Secondly, the outcome for the horse after racing was recorded. Outcomes were broadly categorised as (1) performance, whereby the horse was used for equestrian performance disciplines; (2) pleasure, whereby the horse was being used as a pet, companion horse, or for low-level athletic activities, or (3) breeding, whereby the horse was being used for breeding purposes. Performance horses were further categorised into (1) eventing/show-jumping, whereby the horse was used as an eventing horse, performing in cross-country, show-jumping and dressage or for show-jumping alone; (2) show horse/dressage, whereby the horse was being used to perform flat work in the show ring and/or the dressage arena, or (3) pony club/all-rounder, whereby the horse was being used in any combination of activities including flatwork, sporting, mustering cattle and jumping.

### 2.9. Data Analysis

Data analysis was performed using Stata 15.1^®^ (Statacorp, College Station, TX, USA). Histograms were used to assess continuous data (age and racing and performance characteristics) for normality. Normally distributed data were presented as mean and standard deviation. Non-normally distributed data were presented as median, interquartile range unless otherwise indicated. The incidence of retirement was reported as weekly percentages. The weekly percentages of retirements were calculated by dividing the number of new retirements by the number of horses in training (and therefore at risk of retirement) each week multiplied by 100. The median percentages of retirements per week at risk over the 56 weeks were then calculated.

Poisson regression was used to assess the effect of training track and season on the number (counts) of retirements. The exposure variable showing the total count of horses at risk each week was used as the offset to determine our incidence rate ratio (IRR). A clustered model was used to adjust for correlations between horses within the same trainer. Goodness of fit was tested using post-estimation deviance statistics and Pearson statistics. Negative binomial regression models were subsequently used because the post-estimation deviance statistics and Pearson dispersion value were significant, indicating overdispersion of data. Analysis was performed to provide the IRR and the 95% confidence intervals for counts of retirements each week. Variables were considered for inclusion in the multivariable model if the univariable negative binomial regression model was significant at *p* < 0.2. Multivariable analyses were performed with a manual backward stepwise procedure based on *p*-values, change in odds ratio between successive models and biological plausibility to obtain a parsimonious model.

Logistic regression was used to assess the effect of racing career and performance indices on whether retirement was voluntary or involuntary. Variables were considered in the multivariable model if the univariable logistic regression model was significant at α = 0.2. Multivariable analyses were performed using a backwards stepwise procedure to obtain a parsimonious model. Goodness of fit was confirmed with Pearson’s and Hosmer and Lemeshow’s goodness-of-fit tests. Pearson’s Chi-squared test was used to compare whether retirement was voluntary or involuntary and the outcomes for horses after retirement. A significance at *p* < 0.05 was set for all multivariable models.

## 3. Results

There were forty trainers (15 at Eagle Farm, 6 at Doomben, 12 at Deagon and 7 at both Doomben and Eagle Farm) eligible for enlistment. Twenty-seven trainers (11 at Eagle Farm, 5 at Doomben, 4 at Deagon, and 7 at both Doomben and Eagle Farm) consented to participate (Figure 1).

Data were collected at weekly intervals over a 13 month (56 week) period from November 2017 to December 2018 for 26/27 (96%) of trainers. The one trainer that did not complete the study contributed six months of data before retiring from training and, therefore, had to withdraw from the study.

### 3.1. Incidence of Retirement

There was a median of 544 racehorses in training per week (IQR 538, 547). There was a total of 110 racehorse retirements over the study period, a median of 2 (IQR 1, 3) retirements per week, which equated to 0.4% (IQR 0.2%, 0.6%) of the horses in training per week. Eight horses were excluded from this analysis because they were reported as retired from racing but cross-checking with the RA public database [8] revealed that seven had been transferred to another trainer and one had been exported to China.

### 3.2. Effect of Training Track and Season

While season appeared to influence the rate of retirement, the racetrack did not (Table 1). There was 0.56 times the rate of retirements (a 44% lower rate of retirement) in autumn relative to winter (*p* = 0.02), although the overall effect of season on retirements was not significant (*p* = 0.07). Multivariable analysis was not performed as the season was the only variable eligible for inclusion in a multivariable model.

### 3.3. Reasons for Retirement

The reasons for retirement are summarized in Table 2. Involuntary retirements comprised 56/110 (51%), and voluntary retirements comprised 54/110 (49%) of horses in this study. The largest number of retirements were due to musculoskeletal injuries, 40/110 (36%).

### 3.4. Population Characteristics of Retired Horses

The population characteristics of the retired horses, stratified by whether retirement was involuntary or voluntary, are summarized in Table 3.

### 3.5. Racing and Performance Characteristics of Retired Horses

The racing career and performance history of the 99/110 horses that had raced prior to retirement, stratified by whether retirement was voluntary or involuntary, is summarized in Table 4.

### 3.6. Effect of Population, Career and Performance History on Whether Retirement Was Voluntary

Results from univariable and multivariable logistic regression models evaluating the effect of the population, career and performance characteristics on whether a horse was voluntarily or involuntarily retired are presented in Table 5. Following univariable analysis, the variables eligible for inclusion in the multivariable model were: started as a two-year-old, age at first race, length of career, number of starts, starts per year, total race distance accumulated, average distance per start, percentage of races won and percentage of races 1st, 2nd or 3rd place. Multivariable logistic regression analysis determined that with every incremental increase in the total number of race starts, the odds of voluntary retirement increased by 5% (OR = 1.05; 95% CI 1.02 to 1.10, *p* = 0.01). Additionally, with every additional race start per year of racing, the odds of voluntary retirement increased by 21% (OR = 1.21; 95% CI 1.02 to 1.46, *p* = 0.03). However, with every 1 unit increase in the percentage of first, second and third places awarded, the odds of voluntary retirement decreased by 6% (OR = 0.94; 95% CI 0.91 to 0.97, *p* < 0.001).

### 3.7. Medium-Term Outcome of Horses after Retirement from Racing

We determined the medium-term outcome (median follow-up time 14 months, IQR 11, 18, range 8–21) of horses after retiring from racing. A total of 105 of 110 horses (95%) were alive and under care at the end of the follow-up period. There were five horses (5%) that were dead. Of these five horses, four were euthanased and one horse was sent to an abattoir due to severe illness or injury. One horse was euthanased after sustaining severe injuries in a paddock accident whereby it galloped through a fence one month after retirement. One horse was euthanased five months after retirement following exacerbating the original injury, a midbody sesamoid fracture. One horse was sent to an abattoir five months after retirement following exacerbating the original injury, a basilar sesamoid fracture. One horse was euthanased following a synovial laceration and infection, 8 months after retirement. One horse was euthanased 12 months after retirement due to severe colic.

Overall, 108/110 horses (98%) were successfully repurposed after retirement from racing, and 2/110 horses (2%) were lost to follow up (Table 6). Most horses (50/110; 46%) were repurposed as performance horses. No horses were euthanased or sent to an abattoir by an owner or trainer after retirement from racing. One horse was sold to be used as a performance horse but was recognized at the disposal sales and purchased by the jockey who used to ride him. He was subsequently repurposed as a pleasure horse.

### 3.8. How Whether Retirement Was Voluntary or Involuntary Affected the Medium-Term Outcome after Racing

A summary of how whether retirement was voluntary or involuntary affected the outcomes for the horses is presented in Table 7. Horses that had been voluntarily retired had 2.28 times the odds of being repurposed as performance horses than those that were retired involuntarily (*p* = 0.03). Whether retirement was voluntary or involuntary did not influence whether horses were used for breeding or pleasure.

## 4. Discussion

The incidence and reasons for racehorse retirement in the current study (110 horses; 0.4% of horses in training per week) were low compared to a New Zealand study, whereby 555/1571 (35%) horses exited the study due to retirement [5]. A reason for the low incidence of retirement in the current study may be due to the opportunity for horses performing poorly at metropolitan (city) or provincial (intermediate) level to be sold on to race at the regional (country) level, rather than being retired from racing altogether. Unfortunately, following the outcomes of horses sold to other trainers was beyond the resources of this study. In contrast, other studies have reported a greater number of voluntary retirements due to a lack of ability rather than an injury or medical condition [4,5,6]. It is possible that those horses were retired for their lack of ability because there was no opportunity to sell these horses to race at lower levels. The largest proportion of retirements were due to MSI (40/110; 36%), although there was no significant difference in the number of involuntary (56/110; 51%) and voluntary (54/110; 49%) retirements. Non-catastrophic MSIs are often career-ending, and catastrophic MSIs are the predominant cause of death in thoroughbred racehorses worldwide [9,10,11,12,13]. A better understanding of the risk factors for MSI in this population would reduce the incidence of involuntary retirements.

We could not recommend any interventions or areas for further research into retirement based on the season or racetrack the horse is trained on as risk factors. There were fewer retirements in autumn than in winter, although the effect of season was not significant overall. This was attributed to the timing of feature races, the breeding season and influx of the new crop of yearlings. Horses are unlikely to be retired in autumn, during the leadup to the winter carnival, which contains Queensland’s major races. Horses are more likely to be retired in winter, after the completion of the carnival, in time for the breeding season, which commences in spring. There is a commercial advantage to breeding mares as early as possible in spring, to produce foals as close to August 1st as possible to gain a competitive advantage for the highly paid two-year-old races. The older horses that have reached their level (lacked the ability to progress to higher classes of races) or have shown insufficient ability are also likely to be retired in winter to make room for the new crop of yearlings entering the racing stables. The training track did not affect the number of retirements in the current study. This may be because the training track did not affect the risk of MSI in this population of horses [7], and the largest proportion of retirements were due to MSI.

The odds of voluntary rather than involuntary retirement increased by 5% with every additional race start and by 21% for every additional race start per year of racing. This was attributed to survival bias or the healthy horse effect, whereby those horses that are fit to complete a larger number of races are retired for voluntary reasons rather than due to injury [14,15,16]. In contrast, the odds of voluntary rather than involuntary retirement decreased by 6% with every additional increase in the percentage of first, second and third places awarded. It is possible that this may be due to owners seeing the potential for higher future earnings in these horses and continuing to race them despite an impending musculoskeletal injury. A case-control study would further elucidate these risk factors for voluntary versus involuntary retirement.

Follow-up of the medium-term outcomes for horses after retirement from racing revealed that 108/110 (98%) of horses were successfully repurposed after racing, with almost half (50/110; 46%) taking up second careers as performance horses. In the current study, whether retirement was involuntary or voluntary did not affect horses used for breeding or pleasure, whereas performance horses were more likely to have been retired voluntarily. This is intuitive as performance horses must be sound for their new athletic career. However, 12/40 (30%) of the horses that retired due to musculoskeletal injury were successfully repurposed as performance horses. Tendon injuries occur during racing when the loading forces experienced at high speeds exceed the physiological limits of the tendon in the central region [17,18,19]. Similarly, fatigue fractures result from bone microdamage, which forms from either excessive intense forces in well-adapted bone or from a lower level of exercise in poorly adapted bone such as when returning from rest [20,21,22,23,24]. It could be argued that these forces are not experienced with performance horse disciplines and consequently that racehorses may be suitable for a performance athletic career despite their injuries having prevented them from racing.

The main strength of this study was the acquisition of complete, detailed and accurate data due to the unique access to trainers through personal interviews. Relying on trainers to complete standardized questionnaires results has resulted in incomplete and unreliable data [12,25,26]. Frequent data collection avoided recall bias and misclassified data [27,28]. Our weekly contact with trainers also avoided the potential for recall and attrition bias [27,29,30]. Selection bias was minimised by enrolling all horses under the care of trainers rather than allowing trainers to nominate which horses would be discussed [30,31].

The predominant limitation of this study is that it investigated retirement within the Australian racing industry. Whilst most horses were successfully repurposed immediately after racing in this population, our results may not be globally applicable. We also analysed count data to determine our incidence rate of retirement rather than individual horse-time-at risk. Although this method facilitates rapid population-level data collection, it provides a less accurate measure of incidence [27]. However, collecting individual-level data for the entire study population was beyond the available resources, and pilot studies indicated that trainers would not commit the time required for collecting individual horse data. Finally, we only evaluated the medium-term outcome of horses (median follow-up time 14 months, IQR 11, 18, range 8–21). Although these findings suggest that successful repurposing of horses and their welfare was maintained in the medium-term, these results may not reflect the long-term outcome for horses in their new careers. Currently, once racehorses have retired from racing, there is insufficient control over the long-term welfare of these horses [1]. There is a need for traceability and accountability for these horses to ensure that their welfare is maintained in their new careers.

## 5. Conclusions

In conclusion, we reported a low incidence of retirements, and the greatest cause of retirement was musculoskeletal injuries. Season and training track did not affect the risk of retirement. The odds of voluntary rather than involuntary retirement increased with every additional race start and race start per year of racing but decreased with every additional increase in the percentage of first, second and third places awarded. Consequently, we recommend that these risk factors are addressed in future investigations into retirement of racehorses. Ninety-eight per cent of horses were successfully repurposed after racing, with the majority becoming performance horses. Thoroughbred racing attracts significant media and public attention. It is vital that the industry is focused on understanding the risks for voluntary rather than involuntary retirement and optimising the successful long-term repurposing of horses. There is a need for traceability and accountability for these horses to ensure that their welfare is maintained in their new careers.

## Figures and Tables

**Figure 1 animals-11-00142-f001:**
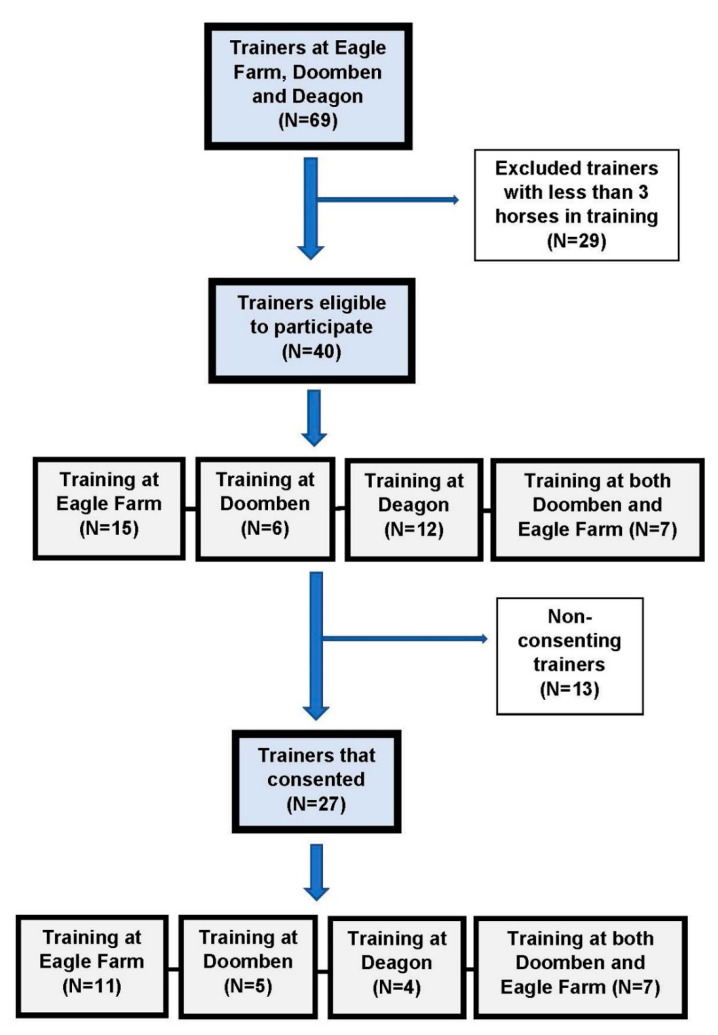
Recruitment of trainers to investigate retirement in thoroughbred racehorses in South East Queensland, Australia. N = the numbers of trainers at recruitment time at the start of November 2017.

**Table 1 animals-11-00142-t001:** The effect of training track and season on the number of retirements in thoroughbred racehorses training in South East Queensland.

Variable	Univariable IRR (95% CI)	*p*-Value
Track		
Eagle Farm	reference	-
Doomben	1.18 (0.56, 2.48)	0.67
Deagon	1.43 (0.70, 2.90)	0.33
Doomben and Eagle Farm	1.16 (0.46, 2.93)	0.75
Season		
Winter	reference	-
Spring	0.63 (0.40, 1.01)	0.06
Summer	0.66 (0.40, 1.10)	0.11
**Autumn**	**0.56 (0.34, 0.91)**	**0.02**

IRR = Incidence Rate Ratio. Univariable negative binomial regression models shown. Variables significant at α = 0.05 indicated in bold.

**Table 2 animals-11-00142-t002:** The reasons for retirement in 110 Thoroughbred racehorses in South East Queensland.

Reason for Retirement	Number (%)
Involuntary retirement	
MSI	40 (36)
Upper respiratory tract	8 (7)
Lower respiratory	6 (5)
Cardiac	1 (1)
Barrier/behaviour problems	1 (1)
Total Involuntary retirements	56 (51)
Voluntary retirement	
Lack of ability	24 (22)
Reached their level	21 (19)
Impending MSI	9 (8)
Total Voluntary retirements	54 (49)
Total Retirements	110 (100)

**Table 3 animals-11-00142-t003:** The population characteristics of 110 retired Thoroughbred racehorses in South East Queensland, stratified by whether retirement was voluntary or involuntary.

Retirements	Age in Years	Sex N (%)			Unraced N (%)		Retired as Two-Year-Old N (%)	
	Mean (sd)	Stallions	Mares	Geldings	Yes	No	Yes	No
Involuntary Retirements	4.9 (1.3)	2 (4)	26 (46)	28 (50)	6 (11)	50 (89)	2 (4)	54 (96)
Voluntary Retirements	5.5 (1.8)	1 (2)	21 (39)	32 (59)	5 (9)	49 (91)	1 (2)	53 (98)
Overall Retirements	5.2 (1.6)	3 (2)	47 (43)	60 (55)	11 (10)	99 (90)	3 (3)	107 (97)

**Table 4 animals-11-00142-t004:** The racing career and performance history of 99 retired horses that had raced prior to retirement, stratified by whether retirement was voluntary or involuntary.

Variable	Involuntary Retirements (N = 50)	Voluntary Retirements (N = 49)	Total Retirements (N = 99)
Started as two-year-old			
Yes	26 (52%)	16 (33%)	42 (42%)
No	24 (48%)	33 (67%)	57 (58%)
If started as two-year-old, number of starts	2 (1, 5)	3 (2, 6)	2 (1, 5)
Age at first race (years)	2 (2, 3)	3 (2, 3)	3 (2, 3)
Length of career (months)	30 (16)	36 (20)	34 (18)
Number of starts	17 (8, 26)	27 (11, 43)	21 (11, 37)
Starts per year	7 (3)	10 (3)	8 (3)
Total race distance accumulated (5 furlongs/1 km)	22 (9, 36)	41 (15, 58)	26 (13, 47)
Average distance per start (furlongs)	7 (6, 7)	7 (6, 7)	7 (6, 7)
Number of races won	3 (1, 5)	3 (1, 5)	3 (1, 5)
% of races won	18 (8, 21)	11 (5, 15)	13 (5, 19)
Number of races 1st, 2nd or 3rd place	8 (3,12)	10 (2, 15)	9 (2, 14)
% of races 1st, 2nd or 3rd place	41 (31, 50)	31 (10, 41)	38 (26, 44)
Total prizemoney ($1000 AUD)	75 (20, 227)	65 (11, 286)	71 (16, 278)
Prizemoney per start ($1000 AUD)	4 (2, 8)	3 (1, 7)	3 (1, 8)

Values reported as either number (percentages), median (interquartile values) or mean (standard deviation).

**Table 5 animals-11-00142-t005:** The effect of population characteristics and the career racing and performance history on whether retirement was voluntary for 110 retired thoroughbred racehorses in South East Queensland.

Variable	Univariable OR (95% CI)	*p*-Value	Adjusted OR (95% CI)	*p*-Value
Age (years)	1.30 (1.01, 1.69)	0.04		
Sex				
Stallions	reference	-		
Mares	1.62 (0.14, 19.07)	0.70		
Geldings	2.29 (0.20, 26.58)	0.50		
Had a race start				
No	reference	-		
Yes	1.18 (0.34, 4.11)	0.80		
**Started as two-year-old**				
Yes	reference	-		
No	**2.23 (0.99, 5.05)**	**0.05**		
Number starts as two-year-old	1.08 (0.83, 1.40)	0.60		
**Age at first race (years)**	**1.81 (0.93, 3.55)**	**0.08**		
**Length of career (months)**	**1.02 (0.99, 1.04)**	**0.19**		
**Number of starts**	**1.04 (1.01, 1.07)**	**0.01**	**1.05 (1.02, 1.10)**	**0.01**
**Starts per year**	**1.30 (1.11, 1.53)**	**0.00**	**1.21 (1.02, 1.46)**	**0.03**
**Total race distance accumulated (5 furlongs/1 km)**	**1.02 (1.00, 1.04)**	**0.02**		
**Average distance per start (furlongs)**	**1.26 (0.90, 1.77)**	**0.18**		
Number of races won	1.04 (0.90, 1.20)	0.60		
**% of races won**	**0.94 (0.90, 0.99)**	**0.02**		
Number of races 1st, 2nd or 3rd place	1.03 (0.97, 1.08)	0.30		
**% of races 1st, 2nd or 3rd place**	**0.96 (0.94, 0.99)**	**0.00**	**0.94 (0.91, 0.97)**	**<0.001**
Total prizemoney ($1000 AUD)	1.00 (1.00, 1.00)	0.30		
Prizemoney per start ($1000 AUD)	0.98 (0.91, 1.04)	0.50		
Retired as two-year-old				
No	reference	-		
Yes	0.51 (0.04, 5.79)	0.60		

Univariable and multivariable logistic regression models shown. In the univariable analyses, variables were considered significant at α = 0.2 and eligible for inclusion in the multivariable model. In the multivariable analysis, variables significant at α = 0.05 were retained in the final model. Significant variables are indicated in bold.

**Table 6 animals-11-00142-t006:** The medium-term outcomes for 110 thoroughbred horses after retirement from racing.

Outcome	Number (%)
Performance	
Eventing/Showjumping	20 (18)
Show Horse/Dressage	14 (13)
Pony club/All-rounder	16 (15)
	50 (46)
Breeding ^†^	41 (37)
Pleasure	17 (15)
Lost to follow up	2 (2)
Total	110 (100)

^†^ A breeding record was confirmed in the Australian Thoroughbred Studbook for all horses except for N = 2, which were intended for breeding at the time of follow-up.

**Table 7 animals-11-00142-t007:** Voluntary and involuntary retirement by the medium-term outcomes for 110 thoroughbred horses after retirement from racing.

Outcome	Performance				Breeding	Pleasure	Lost to Follow up	Total
Reason for Retirement	Eventing/ShowJumping	Show Horse/ Dressage	Pony Club/ All-Rounder	Total Performance				
Involuntary retirement	N (%)	N (%)	N (%)	N (%)	N (%)	N (%)	N (%)	N
MSI	2 (5)	4 (10)	6 (15)	12 (30)	19 (48)	8 (20)	1 (2)	40
Upper respiratory	2 (25)	0 (0)	2 (25)	4 (50)	4 (50)	0 (0)	0 (0)	8
Lower respiratory	2 (33)	1 (17)	0 (0)	3 (50)	1 (17)	2 (33)	0 (0)	6
Cardiac	0 (0)	0 (0)	0 (0)	0 (0)	0 (0)	1 (100)	0 (0)	1
Barrier or behaviour problems	1 (100)	0 (0)	0 (0)	1 (100)	0 (0)	0 (0)	0 (0)	1
Voluntary retirement								
Lack of ability	6 (25)	6 (25)	5 (21)	17 (71)	5 (21)	1 (4)	1 (4)	24
Reached their level	4 (19)	2 (10)	2 (10)	8 (38)	11 (52)	2 (10)	0 (0)	21
Impending MSI	3 (33)	1 (11)	1 (11)	5 (56)	1 (11)	3 (33)	0 (0)	9
Total for Outcome	20 (18)	14 (13)	16 (15)	50 (45)	41 (37)	17 (16)	2 (2)	110

## Data Availability

The data presented in this study are available on request from the corresponding author. The data are not publicly available due to privacy.

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
