# Peer review of "Appraising the Welfare of Thoroughbred Racehorses in Training in Queensland, Australia: The Incidence, Risk Factors and Outcomes for Horses after Retirement from Racing"

_animals, 2021, doi:10.3390/ani11010142_

Round 1

Reviewer 1 Report

Simple Summary and Abstract

Much of the content of these two sections is identical.

Introduction

Lines 80 and 81 – Two sequential sentences beginning with furthermore.

Line 119 – It is quite surprising that ‘most trainers had an excellent understanding of their veterinarians’ diagnoses’.  Was this based on what the trainers reported or what the actual veterinarian’s records reported?

Line 133 – What was the definition used for ‘training’?

Line 138 – It is not clear why injured horses were not captured in the assessment of retirement.

Line 141 – The mention of behavior as a specific aspect of retirement is not mentioned or justified in the introduction or objectives.  Why was behavior deemed important where other non-catastrophic injuries (e.g., colic, stress fracture, desmitis, etc.) were not specifically mentioned?

Line 201 – It is not clear how Poisson regression (log-linear model) was used to assess a continuous variable (number of retirements) versus categorical data (training track and season).

Line 204 – It is not clear what is referred to or the purpose of assessing as ‘known and unknown differences between trainers’.  How does this specifically relate to your objectives?

Line 205 - It is not clear what is referred to ‘Pearson statistics’.  It this Pearson correlation coefficient?

Line 209 – The use of a p-value of <0.2 seems very gratuitous here.

Line 214 – Again, the use of a p-value (alpha) of <0.2 seems very gratuitous here.

Line 218 - Significance was not set at p< 0.05 for all tests – see above two comments.

Line 221 – Why the mention of 50 trainers at Gold Coast, 56 trainers at Sunshine Coast and 4 trainers at Ipswich in Appendix A2 if they were not included in this study?

Line 232 – Appendix A2 mentions 612 horses, but there is no mention of how many horses were included in the study population.

Line 242 – Why was racetrack not included in the multivariant analysis?

Line 245 – How were the reference parameters of Eagle Farm and Winter selected versus the other choices of track and season?

Line 257 – Table 2 needs to have subtotal labels included for the involuntary and voluntary retirement values.  None of the tables in the manuscript have been formatted according to journal guidelines.

Line 266 – Table 3 – The term ‘Entire males’ is not commonly accepted terminology for stallions.  Reorganize and rename sex categories – Mares, Geldings, Stallions - if these terms are to be used.

Line 274 – Need to be consistent in tables – Remove ‘N=’ and ‘%’ in the first 2 rows.  Add footnote that the values are a mixture of mean-standard deviations and median-interquartile values.

Line 292 – Table 5 – Why was ‘No’ used as the reference for ‘Retired as two-year-old’ and ‘Yes’ was used for ‘Unraced’ and ‘Started as two-year-old’?  Be consistent in the significant digits in the p-values throughout the manuscript – 0.6 versus 0.05.

Lines 322 and 323 – Do not know what ‘How Reason…’ or ‘how the reason’ means.

Line 329 – Table 7 – “Performance’ header needs to be justified across all relevant columns below.

Line 338 – The “greater” number of 56 versus 54 does not meet statistical or clinical significance, hence does not support your argument.  Neither does the mention of MSI being the most common cause of retirement.  How do these numbers compare to other studies?

Line 340 – Need to be clear about the distinction better non-catastrophic and catastrophic MSI here.

Line 342 – Need to provide numbers here for point of references for ‘greater number’.

Line 356 – Unclear what ‘reached their level’ refers to.

Line 359 – It is unclear how ‘musculoskeletal injury in this study and the training track’ relates to risk of injury.  Is it one or both?

The repeated mention of ‘the first study’ is quite self-serving and does not provide perspective if this only refers to Australia or to all racing research in general.  It is was not ‘the first’ then what is the value of publishing the paper?

Line 380 – The mentioned mechanism of action for tendon injury may not specifically relate to ligaments.

Line 387 – While the weekly interviews with trainers was a strength, it is not clear that the ‘large number of trainers’ was the main strength of the study as there was no specific assessment of trainers.  The focus was on horses and their performance.  It is not clear how many horses were actually included in the study, so determination if horse numbers was a ‘strength’ cannot be fully assessed.

Line 395 – ‘of this study is that this study’ is poor wording

Lines 411 and 413 – The lack of attention throughout the paper on proper formatting and punctuation detracts from the favorable review of this work.  This is reflected also in the references with regard to mixed capitalization, punctuation and formatting of journal names.

Figure 2A – I do not see the value of this material as it does not specifically relate to the consents of this manuscript.

Reviewer 2 Report

Thank you for the interesting paper, which provides (exp. in combination with the paper about MSIs) a unique and profound insight into animal welfare issues of race horses. Even though I understand, that there are limitations in the man power and fundings of every research project, an individual follow up of the horses, including those sold to other trainers would have been interesting.

Here are a few detailed comments:

L29: I think it should be starts/year of racing and not start/year?

L138/139: Why? At the end of the 13 month observation period it would have been possible and interesting to follow up these horses.

L152: What ist considered to be significant? Did you also record or count behavioural disorders like weaving? 

L231ff: What was the total number of horses in Training (n=?), What were the numbers of horses in Training per week? Please provide descriptive statistics to enable the reader to put the number of 110 retired racehorses into perspective

L337: Was there a follow up on these horses after the end of your 13 month study period? These horses could have retired as well within you study period. 

Reviewer 3 Report

I enjoyed this paper and recognise its potential value. 

I do however have a small number of overarching comments:

  • please define some key terms.
  • reconsider the way that 'outcomes of' is used, this would seem to be better presented as 'outcomes for (the horse)'
  • excessive use of 'we' and 'our' throughout the text
  • avoid using lists - keep to writing in continuous prose in all sections.

->There is NO reference made to Human Ethics approval being secured allowing this study to be conducted using interviews.

->welfare is not covered throughout the text but features strongly in the conclusion. 

I have provided a marked-up copy of the manuscript with track changes and comments used in MS Word. 

I have provided a marked-up copy of the manuscript with track changes and comments used in MS Word. I hope that you find this useful.

Round 2

Reviewer 1 Report

Thank you for the detailed and thoughtful replies to the provide comments and suggested edits.  I appreciate your time and effort in this endeavor.  The manuscript reads much better now, and the content is easier to follow.

Tables – A few of the tables are still not formatted (e.g., centered, justification) according to the journal’s guidelines provided in the template.  Please review and revise as needed.

Table 3 – I recommend reordering the columns to help the reader process the information more readily.  Age > Sex > Unraced > Retired.  This allows a flow or progression in the data reported.

Table 5 – Same comment here about ordering rows chronologically and adding subheaders to demark relative grouping of data.  Right now, it appears to be all one long and seemingly unrelated list of terms and data.

Age > Sex > Starts > Retired, etc.

References – Several of the journal titles still lack consistency in format (e.g., spelled out, abbreviated, periods).

Author Response

Thank you for the help with the tables. I have re-formatted and re-ordered the rows and columns. I agree, they are much easier to follow that way. 

I am having trouble with the references. I had already re-formatted them and have done so again, but I believe that Endnote is re-setting them. I have saved the changes so hopefully that has worked this time. 

Reviewer 3 Report

Very nice to see very nearly all my round 1 review comments have been addressed.  Thank you. 

There is however, just one point that needs to be addressed - no mention still of human ethics approval for carrying out the study and talking to the participants.  This needs to be included in the Methods section as indicated in my Review round 1 comments.

Author Response

Sorry! I had misunderstood and thought that you were seeking confirmation that we had ethics approval to conduct the research rather than asking for it to be written in the methods. This has been amended (line 109)